# Abdominal Photobiomodulation and the Gut-Brain Axis: A Systematic Review of Mechanistic and Translational Evidence

**DOI:** 10.3390/biomedicines13123042

**Published:** 2025-12-11

**Authors:** Gabriela N. F. Guimarães, Fabrizio dos Santos Cardoso, Laura Gamboa, Douglas W. Barrett, F. Gonzalez-Lima

**Affiliations:** 1Texas Consortium in Behavioral Neuroscience, Department of Psychology, University of Texas at Austin, Austin, TX 78712, USA; 2Centro Universitário Redentor/Afya (UniREDENTOR), Itaperuna 28300-000, RJ, Brazil; 3Hospital do Câncer de Muriaé, Fundação Cristiano Varella (FCV), Muriaé 36880-000, MG, Brazil; 4Hospital São Vicente de Paulo (HSVP), Bom Jesus do Itabapoana 28360-000, RJ, Brazil; 5Department of Psychiatry and Behavioral Sciences, University of Texas at Austin, Austin, TX 78712, USA

**Keywords:** photobiomodulation, gut–brain axis, abdominal, mitochondrial bioenergetics, gut microbiota, neuroinflammation

## Abstract

**Background/Objectives:** Bidirectional communication between the gut and brain is central to neurological and psychiatric health, and abdominal photobiomodulation (PBM) has emerged as a promising non-invasive way to modulate this axis by targeting intestinal mitochondria, epithelial integrity, and the microbiota. We systematically reviewed preclinical and clinical evidence on abdominal PBM, alone or in combined protocols, reporting microbiome, metabolic, or neurobehavioral outcomes. **Methods:** Following PRISMA 2020 recommendations, we searched MEDLINE, Scopus, Web of Science, and ScienceDirect through May 2025 for animal and human studies applying PBM to the abdomen and reporting gut-related, metabolic, or brain-related outcomes. **Results:** Nine studies met the eligibility criteria (five human, four animal). Human trials, mainly in Parkinson’s and Alzheimer’s disease, used 630–904 nm light and reported gains in mobility, balance, cognition, and olfaction; one trial also showed microbiota modulation with a decreased Firmicutes:Bacteroidetes ratio. Animal models revealed cognitive improvement, reduced neuroinflammation, dopaminergic neuroprotection, and microbial rebalancing. Mechanistic findings converged on enhanced mitochondrial bioenergetics, redox and anti-inflammatory signaling, vagal activation, and short-chain fatty acid-mediated effects. **Conclusions:** Current evidence, though limited by small samples, heterogeneous dosimetry, combined treatment sites, and few sham-controlled human trials, suggests that abdominal PBM can influence the gut–brain axis through converging mitochondrial, immune, and microbial mechanisms. Adequately powered randomized trials with standardized dosimetry, validated mechanistic biomarkers, and integrative multi-omics analyses are needed to clarify causal pathways and optimize translational applications.

## 1. Introduction

The increasing prevalence of metabolic and neurobehavioral disorders has been linked to alterations in the gut microbiota, a complex ecosystem that plays a critical role in digestion, immune modulation, and neuroendocrine signaling [1,2]. Disruptions in microbial diversity, known as dysbiosis, have been associated with chronic diseases such as obesity, Parkinson’s disease, and Alzheimer’s disease [3,4,5]. Central to this association is the gut–brain axis, a bidirectional pathway that connects the gastrointestinal tract and the central nervous system via neural, immune, and metabolic signals [6,7].

In neurodegenerative conditions, gut-derived inflammatory signals and microbial metabolites can cross the blood–brain barrier and contribute to neuronal damage [8,9]. In Parkinson’s disease, intestinal dysbiosis and elevated lipopolysaccharides have been implicated in alpha-synuclein aggregation in the enteric nervous system, which may propagate to the brain via the vagus nerve [3,10]. Similarly, in Alzheimer’s disease, chronic infections such as *Helicobacter pylori* and *Porphyromonas gingivalis* have been associated with systemic inflammation and cognitive decline [11,12].

Photobiomodulation (PBM) is a non-invasive light therapy that uses red or near-infrared light to enhance mitochondrial function and reduce inflammation [13,14,15,16]. When applied to the abdomen, PBM may modulate the gut microbiota, improve barrier integrity, and influence brain function through gut-mediated pathways [17,18,19]. Studies suggest that PBM increases beneficial bacteria and reduces pro-inflammatory markers, contributing to the emerging concept of photobiomics [17,20,21]. In clinical practice, however, abdominal PBM is rarely delivered in isolation; most neuromodulation protocols combine abdominal, cervical and transcranial sites. Accordingly, this review focuses on PBM protocols that include an abdominal component, with or without additional targets, when gut-related or neurobehavioral outcomes were reported.

Given the novelty of this therapeutic strategy, this review aims to synthesize current evidence on abdominal PBM and its effects on the gut–brain axis, highlighting mechanistic pathways and therapeutic implications for metabolic and neuropsychological disorders.

## 2. Methods

### 2.1. Data Sources and Search Strategy

A systematic literature review was conducted following the Preferred Reporting Items for Systematic Reviews and Meta-Analyses (PRISMA) guidelines [22,23]. A comprehensive search strategy was applied between January and May 2025 to identify relevant studies of abdominal PBM, either as a standalone intervention or together with other PBM targets, and the effects on clinical outcomes and gut microbiota. Searches were run in MEDLINE (PubMed), Scopus (Elsevier), ScienceDirect (Elsevier), and Web of Science (Clarivate). Final search dates: Pubmed [15 May 2025]; Scopus [15 May 2025]; ScienceDirect [15 May 2025]; Web of Science [16 May 2025]. Reference lists of included articles were also screened.

For MEDLINE, the search query was: (“Photobiomodulation” OR “low-level light therapy” OR “PBM” OR “LLLT”) AND (“abdomen” OR “abdominal” OR “abdominal application” OR “gut-brain axis” OR “enteric nervous system”) AND (“microbiome” OR “microbiota” OR “gut” OR “intestinal flora” OR “microbial” OR “gut bacteria”). Equivalent search strategies were adapted for Scopus, ScienceDirect, and Web of Science to ensure comprehensive coverage. The complete search strings, filters, and logic are provided in Appendix A.

### 2.2. Eligibility Criteria

Inclusion criteria comprised randomized and non-randomized clinical trials, retrospective and prospective studies, case series, and case reports with full-text availability, published in English, Spanish, or Portuguese, without date restrictions, and involving either humans or animal models. Studies were considered eligible if they applied abdominal PBM, and reported at least one outcome related to microbiota composition or neurological and psychiatric symptoms associated with gut–brain axis modulation; we included studies in which the abdomen was one of the PBM target sites, even when combined with cervical or transcranial irradiation, provided that outcomes relevant to the gut–brain axis were reported. Exclusion criteria included ineligible designs such as editorials, letters, commentaries, book chapters, systematic reviews, and narrative reviews, as well as lack of full-text access, publication in other languages, absence of PBM intervention, or outcomes unrelated to microbiota or gut–brain axis involvement.

### 2.3. Data Extraction

Two reviewers independently (G.N.F.G. and L.G.) screened titles and abstracts and full texts for relevance; disagreements were resolved by consensus. Duplicates were removed using Rayyan and manual verification. A standardized table based on the PICO (Population/Patient/Problem, Intervention, Comparison, Outcome) framework was used to systematically collect information, including author, year, country, study design, objectives, key findings, and participant characteristics such as disease profile, intervention type, model, and sample size. Additional details regarding the intervention, outcomes, and main conclusions were also recorded. The process was guided by the Cochrane Handbook for Systematic Reviews of Interventions [24].

### 2.4. Risk of Bias Assessment

The risk of bias was evaluated using instruments appropriate to each study design. The randomized controlled trial [19] was appraised with the RoB 2.0 (Cochrane Risk of Bias 2.0) tool [25], whereas preclinical investigations [18,21,26,27] were assessed using SYRCLE’s Risk of Bias tool [28]. The JBI (Joanna Briggs Institute) checklist [29] was applied to the case series [30], and the CARE (CAse REporting) guidelines [31] were used to evaluate the single case report [17]. Observational and longitudinal non-randomized studies [19,20,32] were analyzed narratively, with bias considered across adapted RoB 2.0 domains.

### 2.5. Certainty of Evidence

The certainty of evidence was assessed using the GRADE framework (Grading of Recommendations, Assessment, Development, and Evaluation) [33] which considers five domains: study design, imprecision, indirectness, inconsistency, and publication bias. Evidence was classified as moderate when all domains were satisfied, moderate when one domain was unmet, low when two were unmet, and very low when three or more were unmet.

### 2.6. Synthesis Methods

Given heterogeneity in designs and outcomes, a pre-specified narrative synthesis was conducted; no meta-analysis or formal heterogeneity/sensitivity analyses were feasible.

### 2.7. Registration and Protocol

Registration: PROSPERO (CRD420251183157). Protocol: Registered protocol in PROSPERO; no amendments were made.

## 3. Results

### 3.1. Study Selection

A total of 254 studies were identified across the databases: MEDLINE (PubMed) (*n* = 86), Scopus (*n* = 118), and Web of Science (WOS) (*n* = 50). After removing 114 duplicates, 140 records remained for screening. Title and abstract screening excluded 14 studies due to ineligible publication type or language, leaving 126 full texts for eligibility assessment. Of these, 117 were excluded for the following reasons: no use of NIR therapeutic window (*n* = 23), abdominal interventions unrelated to the topic (*n* = 14), or outcomes unrelated to the research focus, such as obesity, body contouring, or vascular assessment (*n* = 80). Reports were excluded when PBM was delivered solely outside the red–near-infrared therapeutic window (approximately 630–904 nm); Gordon et al. [27] was retained because abdominal irradiation was performed at 656–670 nm, within the red range of this window, whereas the excluded reports used shorter visible wavelengths or non-therapeutic light exposures. Ultimately, 9 studies fulfilled all inclusion criteria and were included in the qualitative synthesis. The selection process is detailed in Figure 1 (PRISMA 2020 flow diagram), and the corresponding checklist is provided in Appendix A.

### 3.2. Characteristics of Included Studies

The nine studies included five clinical investigations in humans and four preclinical animal experiments. Four animal studies explored abdominal PBM using murine and non-human primate models of neurodegenerative and stress-related disorders. All clinical investigations targeted neurodegenerative disorders, predominantly Parkinson’s disease and Alzheimer’s disease, with a cumulative cohort of 65 participants aged 60–75 years (Table 1). Preclinical studies primarily utilized male C57BL/6 mice (*n* = 5–15 per group) [21,26], Balb/c mice, and primates exposed to 1-methyl-4-phenyl-1,2,3,6-tetrahydropyridine (MPTP) to induce Parkinsonian phenotypes [27]. Treatment regimens ranged from acute, single-session exposures [30] to extended protocols of up to 45 weeks [32].

PBM represented the central therapeutic intervention across all studies, applied to distinct anatomical targets including the abdomen, cervical region, transcranial areas, and distal sites such as hindlimbs [17,18,19,20,21,26,27,30,32]. The spectral range of applied light extended from 630 to 904 nm, with red and near-infrared wavelengths predominating. Considerable variability was observed in the devices employed: three clinical studies utilized the MIDCARE^®^ Class I laser system (Spectro Analytic Irradia AB, Sweden) [17,20,32] which delivers 904 nm GaAs laser radiation via four diodes (30 mW each); two studies adopted the REGEnLIFE RGn530^®^ device (France) [19,30] integrating 660 nm red and 850 nm near-infrared LEDs with a concomitant 200 mT magnetic field; and preclinical investigations implemented custom-built LED arrays or laser probes, delivering irradiation in the 656–850 nm range [18,26,27].

Dosing protocols varied substantially across studies. Several primary reports lacked complete dosimetric information (e.g., beam diameter, irradiance, duty cycle), which limited direct dose comparisons and prevented precise calculation of energy density for some protocols. Clinical interventions most commonly delivered a total energy of 72 J per session (9 abdominal points and 1 cervical point), administered three times per week for 12 weeks [17,20,30], in some cases followed by an extended 33-week phase of home-based treatment [30,32]. Preclinical protocols employed fluences ranging from 10 to 100 J/cm^2^, applied either as single or repeated sessions over 2 to 8 weeks [18,21,26,27]. One study implemented daily PBM exposure for 3 weeks, with 6 min sessions [21].

Clinical outcomes primarily targeted functional and neurocognitive domains. These included assessments of functional mobility (Timed Up and Go, TUG), balance (step test, single-leg stance, tandem stance), cognitive performance (Montreal Cognitive Assessment, MoCA; Trail Making Test B, TMT-B; digit span), as well as olfaction, micrographia, and fine motor skills (spiral test, Nine-Hole Peg Test, NHPT) [17,19,20,30,32]. One clinical investigation focused specifically on gut microbiota alterations, employing 16S rRNA sequencing [17]. Preclinical endpoints encompassed behavioral testing (Y-maze, Novel Object Location Test, NOLT), neuropathological markers (amyloid-β, phosphorylated tau, tyrosine hydroxylase, sirtuin-1), neuroinflammatory indicators (Iba1, GFAP), synaptic density, hippocampal proteomic analyses, and gut microbiota profiling [18,21,26,27].

Across both clinical and experimental studies, PBM exerted convergent effects through multiple mechanisms, including mitochondrial bioenergetic stimulation, modulation of inflammatory pathways, vagal nerve activation, and regulation of gut-derived metabolites and microbial diversity. These mechanistic actions were associated with improvements in cognition, motor function, and microbiota composition, collectively supporting the hypothesis of a gut–brain axis contribution to PBM efficacy [17,18,19,20,21,26,27,30,32].

Geographically, the majority of investigations were conducted in Australia [17,20,27,30,32], with additional studies performed in China [26], France [19], and Spain [21]. All human studies were conducted in outpatient settings, some incorporating home-based treatment protocols, whereas animal studies were performed under standardized experimental conditions in academic biomedical laboratories.

### 3.3. Methodological Quality Assessment

Given the substantial heterogeneity in study designs, the application of a single validated tool for methodological quality assessment was not feasible. The dataset encompassed randomized controlled trials, observational studies, case series, single-patient case reports, and preclinical animal experiments, each of which adheres to distinct methodological standards. While instruments such as the PEDro Scale (Physiotherapy Evidence Database scale) [34] are suitable for randomized trials, they are not applicable to non-randomized, exploratory, or preclinical designs. Consequently, no formal comparative quality score was generated. Instead, methodological rigor was qualitatively assessed within the context of each study type, whereas risk of bias was evaluated separately using appropriate instruments.

To enable cross-study comparison, methodological quality was qualitatively categorized as Low, Moderate, or High, based on four descriptive domains: randomization, blinding, presence of a control group, and justification of sample size. Methodological quality ratings (low, moderate, high) were assigned independently by two reviewers (G.N.F.G. and L.G.) according to these four domains, which are common to established tools such as CONSORT, SYRCLE, and JBI; discrepancies were resolved by discussion with a third reviewer (F.C.S.). These ratings were intended solely as a qualitative summary to facilitate cross-study comparison, not as a formal, validated quality score. Full details are provided in Table 1.

Among the included studies, the randomized controlled trial [19] exhibited the highest methodological quality, with random allocation, allocation concealment, triple blinding, an adequate control group, and an a priori sample size calculation. Four preclinical investigations [18,21,26,27] were rated as high-quality, supported by well-defined protocols, the use of control groups, and blinded outcome assessments, although randomization and allocation concealment were not explicitly reported. Three clinical studies [17,30,32] were classified as moderate quality, owing to the absence of randomization or control groups, despite employing structured follow-up and standardized outcome measures. The single case report [20] was rated as low methodological quality, given its anecdotal nature, lack of comparator, and reliance on subjective reporting. Follow-up durations varied, ranging from two weeks to five years, thereby encompassing both acute and longitudinal assessments.

Despite these limitations, all studies were informative and clearly described eligibility criteria and PBM intervention protocols. Notwithstanding methodological variability, the consistent signals of therapeutic benefit observed across experimental and clinical models strengthen the plausibility of systemic effects mediated by PBM.

### 3.4. Risk of Bias in Included Studies

The randomized controlled trial by Blivet et al. (2022) [19] exhibited low risk of bias across all RoB 2.0 domains, incorporating rigorous methodological safeguards such as random sequence generation, allocation concealment, participant and assessor blinding, complete outcome reporting, and prespecified endpoints. This investigation represents the highest methodological standard within the present dataset.

Among the preclinical studies, assessment with SYRCLE’s Risk of Bias tool indicated low risk in domains related to baseline comparability, housing conditions, uniformity of treatment, and completeness of data [18,21,26,27]. Nonetheless, most studies failed to explicitly report sequence generation or allocation concealment, and only two [21,26] described assessor blinding, raising some concern regarding potential selection and detection biases.

The case series by Liebert et al. (2022) [30], evaluated with the JBI checklist, demonstrated a moderate risk of bias. Strengths included clear eligibility criteria and structured follow-up. However, the absence of assessor blinding, lack of a control group, and reliance on self-reported outcomes increased vulnerability to performance and detection biases.

The case report by Bicknell and Laakso et al. (2022) [17], appraised according to the CARE guidelines, was associated with serious risk of bias, inherent to the anecdotal nature of single-case evidence. The absence of a comparator and lack of independent outcome verification, despite a detailed clinical description and follow-up, limited the generalizability of its findings. Both the retrospective observational study [20] and the non-randomized longitudinal follow-up study [32] were judged to have moderate-to-high risk of bias, primarily due to the absence of randomization, lack of blinding, incomplete outcome reporting, and potential selective reporting. Although these investigations yielded valuable exploratory insights, their methodological constraints compromised internal validity.

Overall, most studies demonstrated low risk of bias in domains pertaining to intervention delivery and adherence to prespecified protocols. However, concerns persisted regarding random sequence generation, assessor blinding, and completeness of outcome data, particularly in non-randomized and single-arm designs. These findings underscore the necessity of methodological standardization and transparent reporting in future PBM research, especially in studies investigating gut–brain axis modulation (Figure 2).

### 3.5. GRADE Evaluation of PBM Studies

The GRADE assessment indicates overall moderate certainty of evidence supporting PBM efficacy in enhancing cognitive performance, attenuating neuroinflammation, and modulating the gut–brain axis.

Cognitive outcomes were evaluated in five studies (subject *n* = 127, including humans and animal models) [19,21,26,30,32]. These studies consistently demonstrated improvements in memory, executive function, and behavioral performance. The certainty of evidence was rated as moderate, supported by low risk of bias in the randomized controlled trial [19] and in two blinded preclinical studies [21,26]. Collectively, these findings justify a favorable recommendation for PBM as an adjunctive strategy to improve cognition in neurodegenerative and stress-related conditions.

Neuroinflammatory markers (e.g., Iba1, GFAP, Sirt1, cytokines) were investigated in four preclinical studies [18,21,26,27], consistently showing reductions in neuroinflammatory expression and restoration of mitochondrial signaling pathways. Given the low risk of bias and moderate biological plausibility, the evidence supports a moderate recommendation for the anti-inflammatory effects of PBM, particularly in the context of chronic neuroimmune dysregulation.

Gut microbiota composition was examined in five studies [17,18,20,21,26], comprising both human and animal data. All reported favorable shifts in microbial communities, including increased abundance of short-chain fatty acid-producing genera (*Akkermansia*, *Faecalibacterium*, *Roseburia*, *Allobaculum*). Despite heterogeneity in sequencing methodologies, the overall certainty of evidence was graded as moderate, supporting a favorable recommendation for PBM as a modulator of the gut microbiota.

Motor and functional outcomes were assessed in three studies from two independent clinical cohorts, involving Parkinson’s disease (*n* = 19 patients) [20,30,32], with tests reporting improvements in mobility (Timed Up and Go), balance (Step Test, Single-Leg Stance), fine motor skills (Nine-Hole Peg Test), and olfaction. Although these studies lacked randomization, the consistency of results across repeated measures and the absence of adverse effects justify a moderate recommendation, with low-to-moderate certainty due to limited sample size and observational design.

Mitochondrial bioenergetics and tissue oxygenation were indirectly supported by three studies [18,21,26], demonstrating improvements in oxidative metabolism, epithelial barrier integrity, and perfusion parameters. While based on surrogate endpoints, the mechanistic consistency and histological validation support a moderate recommendation for PBM in mitochondrial support.

Taken together, the included studies demonstrated low-to-moderate risk of bias, minimal inconsistency across outcomes, and no major concerns regarding indirectness, imprecision, or publication bias, suggesting that abdominal PBM is a biologically plausible, generally safe, non-pharmacological intervention that may improve cognition, attenuate neuroinflammation, and favorably modulate the gut–brain axis and gut microbiota. However, the certainty of evidence is at best moderate, varying by outcome domain, and is largely constrained by small sample sizes, observational designs, and reliance on preclinical models. We therefore frame our conclusions as cautious recommendations for further investigation rather than definitive clinical guidance (Table 2 and Figure 3).

### 3.6. PBM of the Abdomen: Clinical and Preclinical Evidence of Systemic Physiological Modulation

Nine studies were identified which evaluated the effects of abdominal PBM on the gut–brain axis in both clinical and preclinical contexts. Of these, five investigations involved human participants, including patients with Parkinson’s disease, Alzheimer’s disease, and cancer, whereas four studies employed animal models of neurodegeneration or chronic stress. Owing to the heterogeneity of methodologies, treatment protocols, and outcome measures, the findings are presented in two subsections: clinical studies and preclinical (animal) models.

#### 3.6.1. Clinical Studies

In the five clinical studies with abdominal PBM, the protocols, devices, and patient populations varied considerably, reflecting the exploratory nature of the current literature. The characteristics and main outcomes of these clinical studies are summarized in Appendix A.

In a prospective case-series study, Liebert et al. (2022) [30] evaluated the effects of remote PBM in seven males with idiopathic Parkinson’s disease, Hoehn and Yahr stages I–III, all receiving stable anti-Parkinsonian medication. The intervention consisted of 12 weeks of in-clinic PBM three times per week using a class 1 four-diode 904 nm laser applied transcutaneously to nine abdominal points and the C1/C2 cervical region (72 J total per session), followed by 33 weeks of home-based treatment three times per week with a two-diode 904 nm device delivering the same per-session dose. Outcomes included functional mobility (Timed Up and Go as the primary measure), additional mobility tests, dynamic and static balance, fine motor control, cognition, micrographia, and olfactory function, assessed at enrolment, after 12 weeks, and after 45 weeks. The study found improvements in median Timed Up and Go, walking speed, dynamic balance, fine motor performance, and Montreal Cognitive Assessment scores after the clinical phase, with many gains in mobility and cognition maintained at 45 weeks despite some deterioration in other measures. Two participants showed objective improvement from total anosmia to severe microsmia, and no adverse effects were reported, with carers confirming adherence to the home protocol.

In a pilot randomized, double-blind, sham-controlled trial, Blivet et al. (2022) [19] investigated the safety and cognitive effects of PBM in 53 patients with mild to moderate Alzheimer’s disease (mean age 73 years). Participants received daily 25 min sessions, five days per week for eight weeks, using a system combining a cranial helmet and an abdominal belt equipped with pulsed LEDs (850 nm), red LEDs (660 nm), and near-infrared LEDs (850 nm), along with a static magnetic field pulsed at 10 Hz. The primary endpoint was the Alzheimer’s Disease Assessment Scale-Cognitive Subscale (ADAS-Cog). While no significant differences were observed in the total ADAS-Cog score, the PBM group demonstrated selective improvements in comprehension and executive function, with no significant adverse events reported.

Bicknell and Laakso et al. (2022) [17] presented a single-patient case report involving a 57-year-old woman with grade 1 obesity and ductal breast cancer. The patient received abdominal PBM (904 nm, 700 Hz pulse) delivered by a 12-diode laser array, three times per week for 11 weeks. Stool samples were collected before radiotherapy, after treatment, and one-year post-radiotherapy following PBM. The intervention was associated with restoration of microbiota diversity (increased alpha diversity), enrichment of beneficial genera (*Akkermansia*, *Faecalibacterium*, *Roseburia*), and reduction in the number of potentially pathogenic taxa. The patient also reported improvement in joint pain during PBM treatment.

In a prospective proof-of-concept study with five-year longitudinal follow-up, Liebert et al. (2021, 2024) [32,35] evaluated the effects of combined transcranial and remote photobiomodulation in 12 patients with idiopathic Parkinson’s disease, Hoehn and Yahr stages I–III, all on stable antiparkinsonian medication. The protocol used class-1 904 nm lasers over nine abdominal points and the C1/C2 cervical region together with transcranial and intranasal LED devices, delivered in-clinic for 12 weeks (three sessions per week tapered to one per week) and then continued as home-based treatment three times per week for up to five years. Outcomes included mobility and gait (Timed Up and Go, dual-task TUG variants, 10 m walk speed and stride length), dynamic and static balance (step test, tandem and single-leg stance), fine motor performance (spiral drawing, pegboard), and cognition (Montreal Cognitive Assessment), with the five-year follow-up adding MDS-UPDRS-III motor scores, quality of life (PDQ-39) and sleep quality (PDSS). The initial one-year report showed significant and clinically meaningful improvements in mobility, dynamic balance, fine motor skills and cognition that were maintained during the first year of home treatment, with no safety concerns. At five years, six participants who had continued PBM (excluding one with multisystem atrophy and one who stopped treatment) still demonstrated faster gait, longer stride length, better Timed Up and Go performance, improved step tests and higher cognitive scores compared with baseline, while median MDS-UPDRS-III remained stable and no adverse events were reported. Because both articles reported overlapping clinical outcomes in the same cohort, this review extracted quantitative clinical data only from the five-year follow-up paper, using the original proof-of-concept report qualitatively to confirm the consistency and durability of the short-term effects.

In a retrospective analysis of the same cohort, Bicknell et al. (2022) [20] examined gut microbiome changes in 12 patients with idiopathic Parkinson’s disease (mean age 70.8 years) after a 12-week course of PBM to the abdomen, neck, head and intranasal cavity, delivered three times per week for four weeks, twice weekly for four weeks, and once weekly for a further four weeks, using the same four-diode 904 nm laser protocol. Stool samples collected before and after treatment underwent 16S rRNA sequencing, with analysis of alpha and beta diversity, phylum- and genus-level composition, and the Firmicutes:Bacteroidetes ratio. The authors found no statistically significant shifts in overall community structure, but reported a decrease in the mean Firmicutes:Bacteroidetes ratio from 4.60 to 1.58 in 9 of 12 participants, along with trends toward increased abundance of several Bacteroidetes and short-chain fatty acid-producing genera and decreased abundance of selected potentially pathogenic taxa. Alpha and beta diversity metrics did not reach statistical significance and clinical outcomes were not evaluated in this study. No additional safety issues emerged beyond the absence of adverse events already documented in the parent clinical trial. This microbiome study was therefore included separately in the systematic review as a complementary report providing unique gut-related outcomes from the same photobiomodulation protocol, without duplicating the clinical efficacy data extracted from the five-year follow-up paper.

#### 3.6.2. Preclinical Studies

Four preclinical studies investigated abdominal PBM in murine and non-human primate models of neurodegenerative and stress-related disorders. The experimental designs, treatment protocols, and principal findings from these studies are summarized in Appendix A.

In a controlled experimental study, Bicknell et al. (2019) [18] investigated the effects of abdominal PBM on gut microbiota in healthy mice. Animals were allocated to sham, single-exposure, or multiple-exposure groups, receiving laser irradiation at either 660 nm or 808 nm (10 J/cm^2^, 250 Hz). Light was applied to the shaved abdomen via direct contact lens (spot size 0.8 cm^2^), three times per week for two weeks. Fecal samples collected on days 0, 7, and 14 were analyzed by 16S rRNA sequencing, revealing significant microbial shifts. Notably, PBM at 808 nm increased the relative abundance of *Allobaculum*, suggesting wavelength-specific modulation of microbial ecology or greater tissue penetration of photons from a near-infrared laser relative to a red laser [36].

Chen et al. (2021) [26] conducted an 8-week PBM intervention in a murine model with intracerebral injection of amyloid-β (Aβ1–42). PBM was delivered to the upper abdomen using LED arrays emitting at 630, 730, or 850 nm (100 J/cm^2^, 10 mW/cm^2^, 1000 s per session), five times per week. Cognitive performance was assessed with the Morris water maze, alongside immunofluorescence for amyloid and tau pathology, hippocampal proteomics, and fecal microbiota sequencing. PBM-treated animals demonstrated improved memory, reduced plaque burden, downregulation of inflammatory signaling, and restoration of gut microbial balance, including decreased *Helicobacter* and increased *Rikenella*.

Gordon et al. (2023) [27] compared abdominal, leg, and transcranial PBM in both MPTP-treated macaques and mice as models of Parkinson’s disease. PBM (670 nm in macaques, 656 nm in mice; 50 mW/cm^2^, 9 J/cm^2^ for head/abdomen; 4.5 J/cm^2^ for legs) was administered daily for 5 days (primates) or 21 days (mice). Neuroprotection was evaluated by clinical scoring, vertical pole performance, and histological analysis of tyrosine hydroxylase-positive (TH+) and Nissl-stained neurons in the substantia nigra and striatum. When using red lasers, abdominal PBM preserved ~50% more dopaminergic neurons in primates compared to untreated controls and provided superior protection relative to transcranial PBM.

Sancho-Balsells et al. (2024) [21] employed a chronic unpredictable mild stress paradigm to induce depressive-like behavior in mice. PBM (630/850 nm, pulsed 10 Hz, 6 min/day for 3 weeks) was applied via a mixed laser–LED system to the cranium, abdomen, or both, in the presence of a 200 mT magnetic field. Behavioral assessments (Y-maze, novel object location, forced swim test), immunohistochemistry, and microbiota profiling revealed that combined brain–gut PBM elicited the most robust effects. These included enhanced cognitive performance, normalization of hippocampal Sirt1 expression, reductions in IL-6 and TNF-α, and reversal of dysbiosis with increased abundance of *Roseburia*.

Collectively, these preclinical investigations demonstrate that abdominal PBM exerts systemic and neuroprotective effects through multiple mechanisms, including modulation of gut microbiota composition, restoration of microbial diversity, attenuation of systemic and neural inflammation, and preservation of neuronal integrity. Despite variability in wavelength, treatment duration, and anatomical targets, the findings consistently support abdominal PBM as a promising non-invasive intervention with the capacity to influence brain function via gut-mediated pathways. These results provide an experimental foundation for mechanistic discussions and support the hypothesis that abdominal PBM may serve as a neuromodulatory tool within the gut–brain axis.

## 4. Discussion

The current evidence indicates that abdominal PBM can modulate gut physiology and the gut–brain axis through multiple converging mechanisms. To elucidate its systemic effects, we considered the impact of PBM across distinct biological levels. Accordingly, the present discussion is organized into four mechanistic domains: (1) direct effects of PBM on intestinal epithelial cells, including mitochondrial activation and transcriptional regulation; (2) indirect modulation of gut microbiota composition through host-mediated metabolic, immune, and vascular changes, conceptualized within the framework of photobiomics; (3) role of PBM in stimulating mitochondrial biogenesis and enhancing oxidative metabolism in gastrointestinal tissues; and (4) downstream clinical implications of these processes, particularly regarding systemic inflammation, microbial translocation, and neuroimmune interactions that influence brain function and disease. Collectively, these interdependent pathways provide a plausible mechanistic basis for how light applied to the abdominal region can exert remote physiological benefits, supporting its therapeutic potential in gastrointestinal and neuropsychiatric disorders.

### 4.1. Direct Effects of PBM on Intestinal Cellular Physiology

PBM exerts mitochondrial effects through direct photo-oxidation of cytochrome c oxidase (CCO), enhancing electron transport, ATP synthesis, and transient redox signaling [14,16]. Although most mechanistic in vivo studies have focused on the brain and eye, emerging evidence indicates that similar mitochondrial and transcriptional pathways are also active in the intestinal epithelium [37,38,39]. Enhanced mitochondrial respiration in intestinal epithelial cells supports energy metabolism and reinforces the structural and functional integrity of the mucosal barrier [37,39,40,41].

In an experimental colitis model, PBM (630 nm) has been shown to restore crypt architecture, promote epithelial proliferation, and reduce mucosal inflammation [41]. These effects were associated with decreased epithelial apoptosis, preservation of tight junction proteins (e.g., claudin-1), and upregulation of Annexin A1, a key mediator of mucosal repair and resolution of inflammation [42]. At the molecular level, PBM activates redox-sensitive transcription factors such as Nrf2, leading to the upregulation of antioxidant enzymes including superoxide dismutase (SOD), catalase (CAT), and glutathione peroxidase (GPx) [43,44,45,46]. The increase in ATP availability is fundamental for tight junction assembly and mucin production, both of which are essential for maintaining epithelial barrier integrity and limiting bacterial translocation [47,48]. Moreover, PBM has been reported to induce the expression of growth factors such as TGF-β and VEGF, thereby supporting mucosal regeneration, angiogenesis, and vascular perfusion [37,49,50].

### 4.2. Indirect Modulation of the Gut Microbiota: The Concept of Photobiomics

Photobiomics is the emerging science describing how light, through photobiomodulation, indirectly reshapes the gut microbiota by altering host mitochondrial metabolism, redox balance, vascular tone, and immune signaling [51]. By attenuating epithelial reactive oxygen species (ROS) and lactate accumulation while improving mitochondrial efficiency, PBM reshapes luminal nutrient and redox conditions and has been linked to enrichment of short-chain fatty acid (SCFA)-producing taxa such as *Akkermansia*, *Faecalibacterium*, and *Roseburia* [17,18,20,26]. Concomitantly, PBM dampens mucosal pro-inflammatory cytokines (IL-6, TNF-α, IL-1β) and biases macrophage polarization toward an IL-10-producing M2 phenotype, reducing selective pressure for Proteobacteria and favoring eubiosis [20,38,40,52]. PBM-triggered nitric-oxide release enhances microvascular perfusion and oxygen delivery, limiting hypoxia-driven expansion of facultative anaerobes and supporting oxygen-tolerant commensals [53,54,55]. In addition, mitochondrial redox signaling may entrain epithelial and immune circadian programs, stabilizing microbial rhythmicity [56,57]. Collectively, these host-centric effects provide a coherent framework for the indirect modulation of the gut microbiome by PBM.

### 4.3. Effects on Intestinal Mitochondria: Biogenesis and Oxidative Metabolism

Beyond acute activation, PBM appears to drive durable mitochondrial remodeling in the gut, including biogenesis governed by the AMPK–PGC-1α–TFAM axis, which coordinates mtDNA replication, respiratory complex assembly, and cristae reorganization [58]. In injury models, PBM increases mitochondrial membrane potential, augments oxidative phosphorylation, and lowers markers of dysfunction [14,59]. Given that mitochondrial impairment contributes to IBD (Inflammatory Bowel Disease) pathophysiology, barrier failure, and immune dysregulation [60,61], these adaptations are highly relevant to the intestine. For example, enhanced epithelial bioenergetics stabilizes tight junctions, constrains apoptosis, and restores antimicrobial peptide production, while in macrophages and dendritic cells a shift toward oxidative metabolism supports anti-inflammatory polarization and cytokine control [62,63,64]. Collectively, PBM-induced mitochondrial rewiring promotes tissue-wide metabolic reprogramming, from glycolytic inflammation to oxidative resolution and repair, with implications for gut–brain axis homeostasis.

### 4.4. Clinical Relevance: Gut–Brain Axis and Systemic Implications

Abdominal PBM may modulate systemic physiology through the gut–brain axis by jointly improving barrier integrity, mitochondrial efficiency, and microbiome composition. Restoration of the epithelial barrier reduces LPS (lipopolysaccharide) leakage and downregulates Toll-like Receptors (TLR)-dependent immune activation [41]. In rodent models of chronic stress and intracerebral amyloid deposition, abdominal PBM has been shown to reverse behavioral deficits, dampen hippocampal inflammation, and concurrently realign the microbiota [21,26,51]. Mechanistically, enrichment of SCFA-producing taxa may elevate butyrate, which crosses the blood–brain barrier, reprograms microglia, and promotes histone acetylation and neurogenesis [65].

Preliminary human data are consistent with this translational model. In Parkinson’s disease, abdominal PBM is associated with improvements in motor function, cognition, and balance, alongside normalization of the Firmicutes:Bacteroidetes ratio [17,20]. In Alzheimer’s disease, selective gains in executive function and verbal memory have been reported, pending confirmation in larger trials [19]. Importantly, most clinical protocols in Parkinson’s disease relied on 904 nm superpulsed lasers, whereas preclinical studies used continuous red or near-infrared LEDs at 630–850 nm; this divergence in wavelength, pulsing mode and fluence complicates dose–response comparisons and prevents direct extrapolation of preclinical dosing regimens to humans. Taken together, these findings support abdominal PBM as a peripheral neuromodulatory strategy that tunes intestinal mitochondria, microbiota, and immune tone to systemically modulate brain function.

A potential source of confounding, however, arises from the fact that most human studies included in this review employed combined photobiomodulation protocols rather than strictly abdominal irradiation. For instance, the Parkinson’s disease cohort reported in Bicknell et al. (2022) [20], Liebert et al. (2022) [30], and Liebert et al. (2024) [32] received PBM at multiple sites, including the abdomen, cervical spine, transcranial region, and intranasal cavity, while Blivet et al. (2022) [19] used a dual helmet–abdominal belt device in Alzheimer’s disease. Consequently, it is not possible to attribute the observed neurological improvements solely to abdominal stimulation, as contributions from direct cranial or cervical irradiation cannot be excluded. Nonetheless, these studies were retained because they uniquely demonstrated concomitant modulation of the intestinal microbiota, an outcome that cannot be accounted for by transcranial or cervical PBM alone. Moreover, light scattering from cranial applications could still result in indirect abdominal exposure, further blurring mechanistic boundaries. Our goal was therefore not to isolate abdominal PBM as an exclusive intervention but to systematically compile all indexed evidence describing abdominal or multi-site protocols that included an abdominal component, to contextualize the emerging gut–brain photobiomodulation framework.

### 4.5. Mechanistic Plausibility

While the reviewed evidence provides a coherent mechanistic narrative linking mitochondrial activation, immune modulation, and microbial restructuring to brain outcomes, the causal pathway connecting abdominal PBM to neurological effects remains hypothetical. Most current findings are correlational, demonstrating parallel improvements in gut and brain parameters, rather than establishing direct mechanistic causality. Although animal studies have confirmed biological plausibility, no investigation has yet traced PBM-induced molecular events in the gut to specific neural or behavioral outcomes using causal mediation, targeted biomarker tracing, or neuroimaging.

Because our synthesis intentionally integrates preclinical and clinical evidence, mechanistic inferences must therefore be interpreted with caution. Animal models allow invasive readouts of mitochondrial function, neuroinflammation, and microbiota composition that are not yet feasible in patients, whereas human studies provide only behavioral and limited microbiome end points. Thus, converging signals across species increase biological plausibility but do not imply that effect sizes or causal pathways are directly comparable, and we therefore treat animal and human data as complementary strands of evidence rather than as a single homogeneous pool.

Future studies should therefore move beyond association by employing multi-tiered experimental designs that integrate validated mechanistic biomarkers, mitochondrial and immune readouts, and longitudinal assessments of gut–brain communication. Future PBM trials should also adhere to minimum reporting standards, including wavelength, output power, irradiance at the tissue surface, beam geometry, pulsing parameters, fluence per session, and cumulative dose, to enable meaningful cross-study comparison and dose–response modeling. The adoption of standardized dosimetry, sham-controlled protocols, and multi-omics platforms will be essential to elucidate the causal architecture of abdominal PBM and to differentiate systemic photobiological effects from local or confounded contributions. Such an approach will ultimately help clarify whether the gut acts merely as a mediator or as a primary target in the systemic effects of photobiomodulation.

## 5. Conclusions

Abdominal PBM emerges as a promising, non-invasive intervention that can modulate intestinal mitochondrial function, restore epithelial barrier integrity, and indirectly reshape the gut microbial ecosystem. By enhancing oxidative metabolism, engaging redox-sensitive transcriptional programs, and improving tissue perfusion, PBM fosters a local milieu conducive to anti-inflammatory signaling and microbial homeostasis. Converging preclinical and early clinical studies suggest that these local effects translate systemically via the gut–brain axis, with improvements reported in cognition, neuroinflammation, and behavioral outcomes across multiple disease models.

Despite these studies, the evidence base remains constrained by heterogeneity in experimental designs, wavelengths, dosimetry, treatment sites, and outcome measures, as well as small sample sizes and limited sham-controlled human trials. Even so, the recurring mechanistic pattern, linking bioenergetic and immune metabolic modulation in the gut to brain-relevant outcomes, supports further translational testing. Priorities for future work include adequately powered, sham-controlled randomized trials; standardized PBM parameters (wavelength, irradiance, fluence, duty cycle, treatment geometry and schedule); target-engagement biomarkers (mitochondrial and hemodynamic readouts, epithelial barrier integrity, SCFAs, cytokines, perfusion metrics); rigorous microbiome multi-omics; prespecified clinical endpoints; and systematic safety reporting. Integrating PBM with current standards of care may offer a tractable pathway to treat conditions at the intersection of metabolism, immunity, and neurobehavioral function.

## Figures and Tables

**Figure 1 biomedicines-13-03042-f001:**
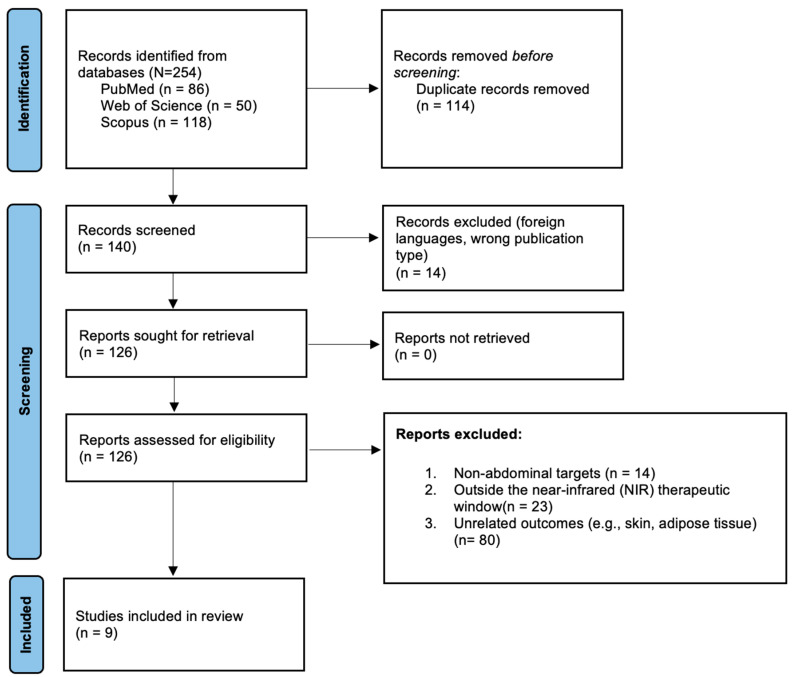
Results of the search and selection process of retrieved records for inclusion in the study. NIR window was defined as red to near-infrared therapeutic wavelengths (630–904 nm).

**Figure 2 biomedicines-13-03042-f002:**
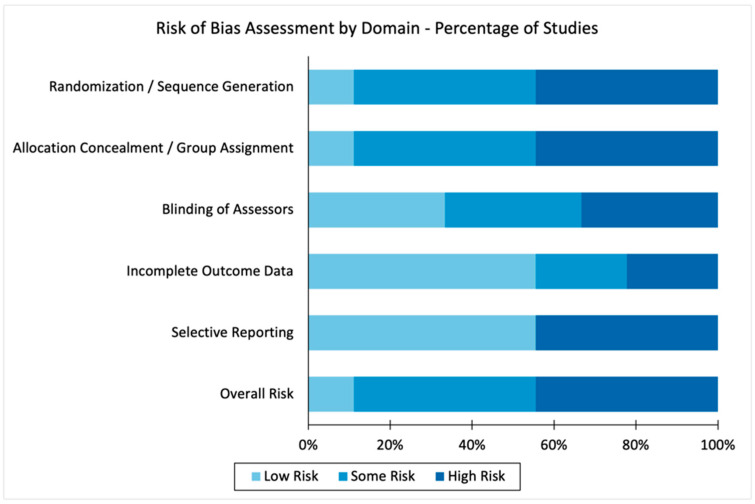
Risk of bias assessment of the included studies. Risk of bias across individual studies according to domains of randomization, allocation concealment, assessor blinding, outcome completeness, and selective reporting. Risk levels were classified as low (light blue, 0), unclear/some concerns (medium blue, 1), or high (dark blue, 2). The overall risk category reflects cumulative judgments across domains for each study.

**Figure 3 biomedicines-13-03042-f003:**
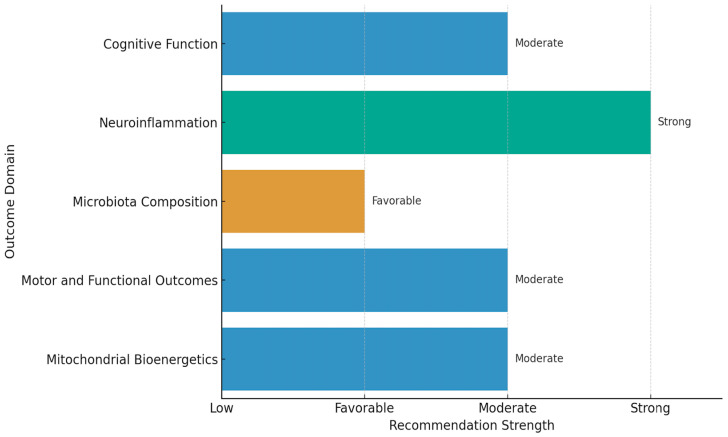
Strength of recommendations for abdominal PBM by outcome domain. Recommendation strength ranged from favorable (microbiota composition) to moderate (cognitive function, motor/functional outcomes, mitochondrial bioenergetics and neuroinflammation). These classifications reflect the integration of evidence certainty, risk of bias, and consistency of findings.

**Table 1 biomedicines-13-03042-t001:** Methodological quality assessment of the included studies.

Author	Design	Randomization	Blinding	Control Group	Sample Justified	Methodological Quality
Bicknell et al. (2019) [18]	Preclinical	Yes (implicit)	Partial (assessor)	Yes	No	Moderate
Chen et al. (2021) [26]	Preclinical	Yes (implicit)	Partial (assessor)	Yes	Yes	High
Bicknell & Liebert et al. (2022) [20]	Observational, retrospective	No	No	No	No	Moderate
Bicknell & Laakso et al. (2022) [17]	Case report	No	Not applicable	No	No	Low
Blivet et al. (2022) [19]	RCT	Yes (described)	Yes (triple-blind)	Yes	Yes	High
Liebert et al. (2022) [30]	Case series	No	No	No	No	Moderate
Gordon et al. (2023) [27]	Preclinical	Yes (implicit)	Partial (assessor)	Yes	No	High
Liebert et al. (2024) [32]	Longitudinal	No	No	No	No	Moderate
Sancho-Balsells et al. (2024) [21]	Preclinical	Yes (implicit)	Partial (assessor)	Yes	No	High

Legend: The table summarizes methodological characteristics of the nine included studies, considering study design, randomization procedures, blinding, presence of a control group, and justification of sample size. Based on these domains, each study was qualitatively classified as low, moderate, or high methodological quality.

**Table 2 biomedicines-13-03042-t002:** GRADE evaluation of PBM across outcome domains.

Outcome Domain	Number of Studies	Human Studies Included	Risk of Bias	Certainty of Evidence	Recommendation
Cognitive function	5	Yes	Low to Moderate	Moderate	Moderate
Neuroinflammation	4	No	Low	Moderate	Moderate
Microbiota composition	5	Yes	Moderate	Moderate	Favorable
Motor and functional outcomes	3	Yes	Moderate	Moderate	Moderate
Mitochondrial bioenergetics	3	No	Moderate	Moderate	Moderate

Legend: The table presents the number of studies per outcome domain, the inclusion of human data, risk of bias, certainty of evidence, and overall recommendation. Certainty of evidence was rated as moderate, with the promising evidence supporting anti-inflammatory effects.

## Data Availability

No new data were created or analyzed in this study. Data sharing is not applicable to this article.

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
