# Peer review of "Abdominal Photobiomodulation and the Gut-Brain Axis: A Systematic Review of Mechanistic and Translational Evidence"

_biomedicines, 2025, doi:10.3390/biomedicines13123042_

Round 1
Reviewer 1 Report
Comments and Suggestions for Authors
Thank you for the opportunity to review this interesting paper that reviews the mechanistic and translational evidence regarding the application of abdominal PBM and its potential effects on the gut–brain axis (GBA).
The content is highly topical presently in the field of photobiomodulation. The authors have taken a contemporary and narrow view of the current field, ensuring the inclusion of abdominal PBM applications which have only become evident in the last 5 years or so in the published literature. This is important when considering the bidirectional aspects of the GBA. I recognise that the authors have also included combined applications of abdominal PBM applications but I believe this is appropriate to do at this early stage of research in this field. Over time, research regarding abdominal-only applications may assist in explicating the mechanistic and translational evidence.
The authors have extracted relevant and accurate information from the cited studies that should assist others to understand what steps should be taken in future to further explicate these factors. This synthesis is exceptionally well structured and reported, and highly valuable.
The authors have identified the relevant papers in the field. They have taken a balanced view and their arguments for including both abdominal-only and combined abdominal + PBM application over other anatomical sites are acceptable. The authors have demonstrated their familiarity with the literature by consolidating the potential mechanisms by which PBM may have an effect on the GBA. Thus, the paper will have significant value at this nascent stage of research in this space.
One could argue that the author conclusions of the methodological quality of the papers (as in Table 1) and GRADE recommendations (as in Table 2) might not be entirely precise but the authors have used the appropriate rating tools to justify their conclusions. Methodological quality can sometimes be in the eye of the beholder. This reviewer notes the expertise of the authorship group and is confident that the content is objectively written with appropriate understanding of the potential mechanisms as examined in the Discussion section.
The tables and figures are otherwise appropriate.
I have reviewed hundreds of manuscripts for many, many journals over some decades. This is the first and only paper that I have ever suggested should be accepted without change/s or corrections of any type – ‘a hole in one’. I heartily congratulate the authors, and highly recommend the publication of the paper in the journal.
Reviewer 2 Report
Comments and Suggestions for Authors
The authors have significantly improved the manuscript. The inclusion of Section 4.4 ('Clinical relevance... A potential source of confounding'), discussing the inability to isolate abdominal effects from cranial effects in human trials, adds necessary scientific rigor. The revision of the GRADE assessment to 'Moderate' instead of 'Strong' is now appropriate and supported by the data. The registration in PROSPERO is also a welcome addition.
- The title 'Abdominal Photobiomodulation and the Gut-Brain Axis' is catchy, but given the honest admission in Section 4.4 that most human studies utilized combined protocols (abdominal + cranial/cervical), consider modifying the title slightly to reflect this nuance, e.g., 'Abdominal Photobiomodulation (Alone or Combined) and the Gut-Brain Axis...'. This would align perfectly with the inclusion criteria stated in the Abstract.
- There is a discrepancy in the level of detail between the clinical and preclinical tables. Table 4 (Animal Studies) provides excellent dosimetric details, explicitly listing irradiance (mW/cm2) and fluence (J/cm2) for most studies (e.g., Chen et al., Gordon et al.)6. However, Table 3 (Clinical Studies) mostly lists treatment duration and frequency without specifying irradiance or fluence. To improve the reproducibility of the review, please add Irradiance (mW/cm2) and Fluence (J/cm2) to the 'PBM Protocol' column in Table 3. If the original papers did not report these, please explicitly mark them as 'NR' (Not Reported). This highlights the need for standardization in clinical reporting.